# Antigen flexibility supports the avidity of hemagglutinin-specific antibodies at low antigen densities

Ananya Benegal[1,2], Yuanyuan He[2,3], Katilyn Ho[2,3], Giselle Groff[2,3], Zijian Guo[2,3], Michael D. Vahey [3,2]*

1 Department of Mechanical Engineering and Materials Science, Washington University in St. Louis, St. Louis, Missouri, United States of America, 2 Center for Biomolecular Condensates, Washington University in St. Louis, St. Louis, Missouri, United States of America, 3 Department of Biomedical Engineering, Washington University in St. Louis, St. Louis, Missouri, United States of America

* mvahey@wustl.edu

## Abstract

The receptor-binding protein of influenza A virus, hemagglutinin (HA), is the most abundant protein on the viral surface. While high densities of HA are thought to improve cellular attachment by increasing avidity for the viral receptor, they may also increase the avidity of neutralizing antibodies. The tradeoff between these two competing effects of avidity is not well understood. To better understand how features of the viral surface influence antibody avidity, we developed fluorescence-based assays to measure dissociation kinetics and steady-state binding of antibodies to intact virions. Focusing on two antibodies that bind to the HA head domain (S139/1 and C05), we confirm that binding orientations that favor bivalent attachment of antibodies to the viral surface can offset weak monovalent affinity by facilitating crosslinking. By modulating HA density in both engineered viruses and synthetic nanoparticles, we find that bivalent antibody binding remains resilient down to one-tenth the HA density on the viral surface and, in the case of C05, that antibody occupancy increases at these lowest densities. Finally, using a combination of structure-guided modeling and antibodies that lock HA in a tilted conformation, we identify flexibility of the HA ectodomain as an additional determinant of antibody avidity. Together, these results establish features of the viral surface that help support or suppress the binding of neutralizing antibodies.

## Author summary

Influenza viruses package high densities of the surface protein hemagglutinin (HA) into the viral membrane during assembly. This promotes viral attachment and entry into host cells by enabling multiple parallel interactions between viral HA and cell surface sialic acid. However, high densities of HA on the viral surface

**Data availability statement:** All relevant data are within the manuscript and its Supporting Information files. Data is also available though Mendeley Data at 10.17632/ryxhs9pgzs.1. Code for analysis is available through GitHub at https://github.com/mvahey/benegal2024.

**Funding:** This work was funded by grants from the National Science Foundation, 2238165 (ANB, MDV) and the National Institutes of Health, AI171445 (YH, ZG, MDV). The funders had no role in study design, data collection and analysis, decision to publish, or preparation of the manuscript.

**Competing interests:** The authors have declared that no competing interests exist.

may also increase viral susceptibility to antibodies whose avidity is enhanced by the availability of multiple HA binding partners. We examined how the density of HA on the viral surface influences the binding of bivalent IgG antibodies that depend on engaging multiple HAs for robust attachment. Using fluorescence-based measurements on authentic and synthetic viruses, we find that reducing HA densities to 10% of their natural levels is not sufficient to disrupt antibody avidity. Additionally, we find that flexibility of the HA anchor may facilitate bivalent antibody binding by allowing HA to tilt relative to the membrane. These findings highlight how both the organization and dynamics of viral surface proteins can influence immune recognition and may inform strategies for antigen presentation in vaccine design.

## Introduction

Influenza A viruses (IAVs) cause seasonal epidemics and occasional pandemics with global mortality in the hundreds of thousands to millions each year [1–4]. Influenza virions are covered with a dense arrangement of the spike proteins hemagglutinin (HA) and neuraminidase (NA). HA, the receptor binding protein, is the most abundant protein on the IAV surface, with hundreds to thousands of copies per virion. The monovalent binding affinity between HA and the viral receptor, sialic acid, is low; as a result, only through multiple simultaneous interactions can a virion stably attach to a host cell [5,6]. The high surface density of HA is thought to play a critical role in facilitating viral binding to the host cell surface during infection by leveraging high binding avidity to compensate for low affinity [7–9] (Fig 1A).

Avidity can also play an antiviral role, by contributing to the binding of neutralizing antibodies. Antibodies against HA are among the most effective protection against influenza virus infection, blocking viral attachment, entry, or release, and engaging with immune cell effector functions [10]. Since natural antibodies contain at least two antigen binding fragments (Fabs), the apparent binding affinity of an antibody can be enhanced through avidity, in some cases by multiple orders of magnitude [11–13]. Antibody crosslinking has also been shown to enhance neutralization through a variety of mechanisms, including viral aggregation and preventing membrane fusion, among others [14–16]. While this effect is widely recognized, the extent to which specific antibodies leverage avidity is shaped by multiple factors and is therefore difficult to predict. These factors could include the antibody isotype, as well as the location, density, and organization of target epitopes on the antigenic surface (Fig 1B). For example, broadly neutralizing antibodies against HA often bind to the more conserved membrane-proximal stalk [17–19]. which is less accessible for bivalent binding [20–22]. In contrast, head-binding antibodies target epitopes that are more sterically accessible, which may be more conducive to bivalent binding [11,12,23]. Understanding how the features of an antigenic surface enhance or suppress antibody avidity could inform the design of next-generation vaccines [24,25].

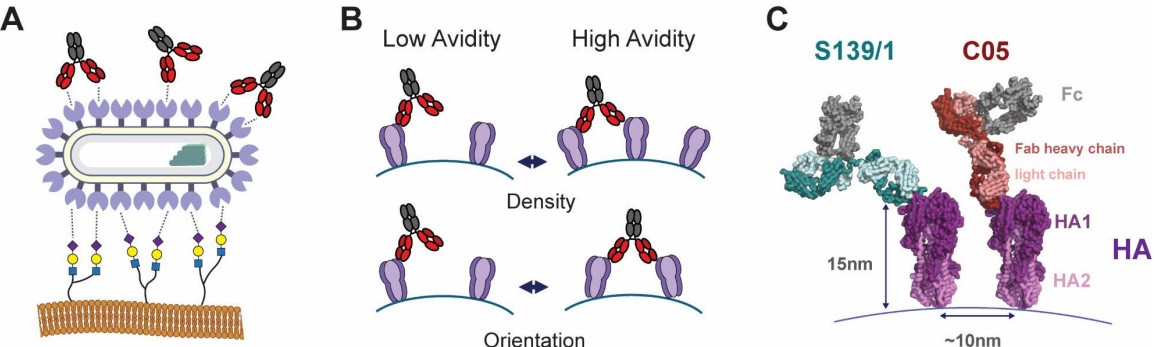

**Fig 1. High densities of HA on the IAV surface can contribute to host cell attachment and the binding of neutralizing antibodies. (A)** HAs on the viral surface bind to multiple sialylated receptors (depicted in the lower part of the image), increasing receptor-binding avidity. High densities of HA also present opportunities for bivalent attachment of antibodies (upper part of the image). **(B)** Factors that may influence the ability of antibodies to bind bivalently are the density of HA on the virion surface and the degree to which they can tilt and rotate. **(C)** Structural model of S139/1 (PDB ID 4GMS, aligned to full-length IgG1 from 1HZH) and C05 (4FQR) bound to the HA head domain (3LZG). Distances indicate the height of the HA ectodomain (~15nm) and typical nearest-neighbor spacing within the viral membrane (~10nm). In all panels, grey portions of antibodies represent the Fc region; colored portions (red for C05, blue-green for S139/1) represent Fabs.

To investigate how the features of antigenic surfaces influence antibody avidity, we developed fluorescence-based assays for measuring antibody dissociation kinetics from intact virions. Using this approach, we determined antigen cross-linking rates to compare the extent to which specific antibodies leverage bivalent binding. Through comparisons of two HA head-binding antibodies – S139/1 [11] and C05 [12] (Fig 1C) – we confirmed that a favorable binding orientation that promotes crosslinking can compensate for weak monovalent affinity. Using these antibodies as models, we tested the effect of decreasing antigen density on antibody binding using virions that contain a fluorescent decoy HA as well as synthetic nanoparticles. We find that both S139/1 and C05 are surprisingly resilient to changes in surface HA density. This resilience depends in part on the ability of HA to tilt extensively about its membrane anchor. Collectively, these findings present a framework for dissecting antibody avidity, and suggest that antigen flexibility – in addition to antigen surface density – may play an outsized role.

## Results

### A fluorescence-based assay to capture antibody crosslinking kinetics

To determine crosslinking rates of HA-specific antibodies, we used a fluorescence microscopy-based approach to measure antibody dissociation kinetics to intact virus particles. Imaging antibody binding to virions allows us to characterize binding kinetics under conditions where HAs are displayed in a physiologically accurate way. Here, we focus on S139/1 [11] and C05 [12], two head-binding antibodies that interact with HA with low monovalent affinity as Fabs but high avidity as bivalent IgG1. We test these against IAV strains of two different HA subtypes; the H1N1 strain A/WSN/1933 (WSN33) and the H3N2 strain A/Hong Kong/1/1968 (HK68). These strains are well-studied, including in the context of these antibodies. Additionally, both antibodies bind HA from these strains with high enough affinity (apparent $K_D$ < 100nM) that they do not require high concentrations in solution, which would increase fluorescence background in our imaging assays.

Both experimental and computational studies have shown that IgG antibodies have considerable flexibility in their hinge region that allow the antibody to sample conformations that promote bivalent binding [26–29]. We model the dissociation of bivalent antibodies as a combination of two rates: 1) the off-rate of a single Fab arm, and 2) the crosslinking rate ('$k_x$') that describes the binding of a free HA by a singly-bound IgG (Fig 2A). This crosslinking rate will depend on the

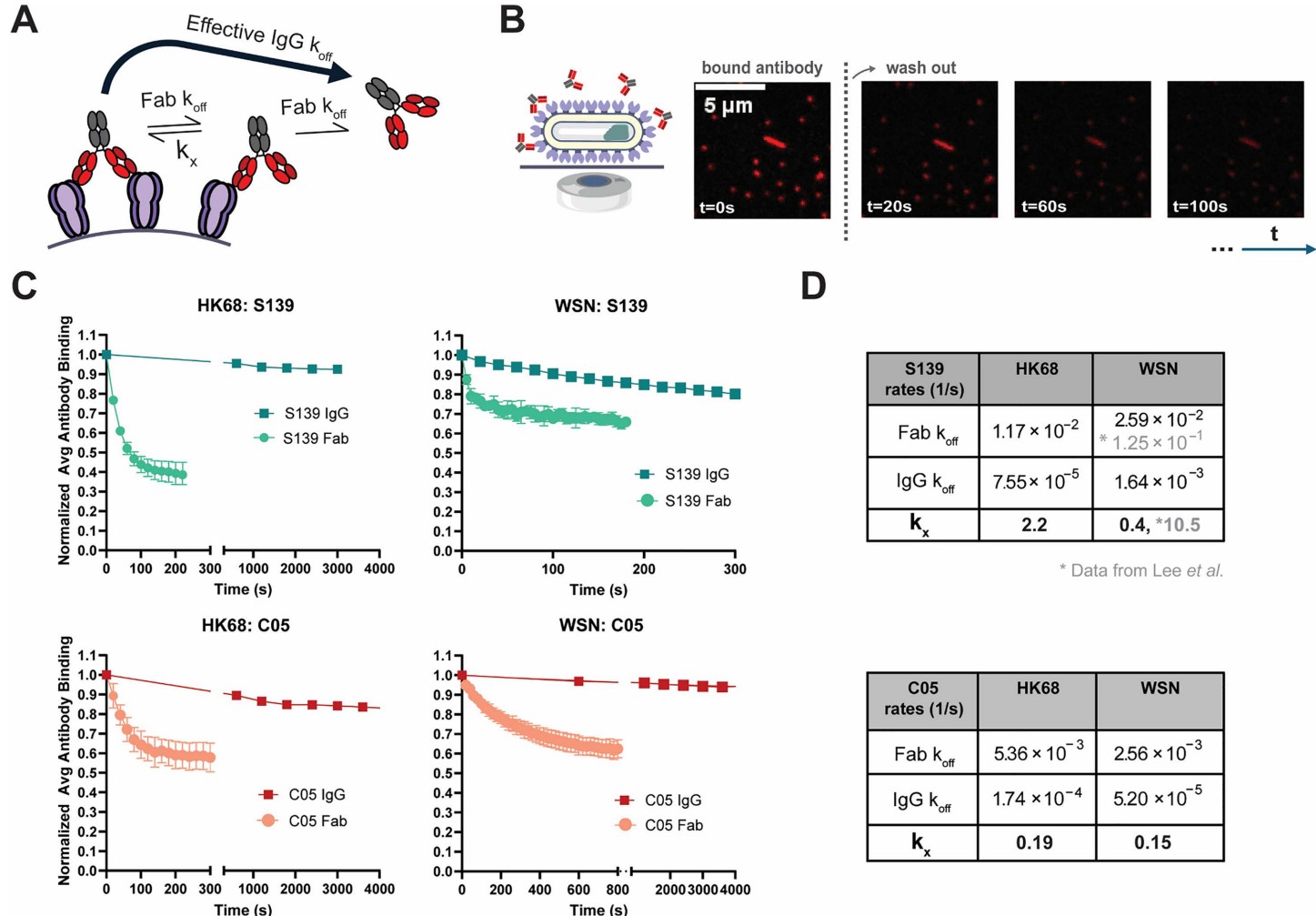

**Fig 2. S139/1 and C05 IgGs are enhanced by avidity to different extents. (A)** The effective off-rate of an IgG antibody (IgG $k_{off}$) is modeled by the combination of two rates: the off-rate for each of the Fab arms (Fab $k_{off}$), and the cross-linking rate ($k_x$) at which the second Fab binds when the first Fab is bound. **(B)** Schematic and images from the assay to measure dissociation kinetics. IAV immobilized onto a flowchamber surface is imaged after equilibrated fluorescent antibody is rapidly washed away. **(C)** Normalized Fab and IgG dissociation curves for both S139/1 and C05 against the HK68 and WSN strains. Data are combined from three biological replicates at two antibody concentrations each and normalized to the fluorescence intensity at t = 0 for each sample. Time between acquisition is set to 5, 20, or 600 seconds depending on relative rate of dissociation. **(D)** Kinetic parameters for S139/1 and C05 Fab/IgG against WSN33 and HK68. The dissociation rates for each antibody/strain combination are determined from the initial rates of fluorescence loss. The crosslinking rate $k_x$ is fit for each pair by simulating the effective IgG $k_{off}$ for a given Fab $k_{off}$ for a range of $k_x$ values. The measured off-rate of S139/1 Fab against WSN33 (0.026 s⁻¹) is likely an underestimate due to low initial signal in this sample and fast dissociation. An estimated $k_x$ value and dissociation rate data from Lee et al. are shown in grey as a comparioson.

association rate of the HA-Fab interaction, the steric accessibility of the epitope, the geometry of binding, and the density of available HAs. To measure these rates, whole virions are immobilized in a flow chamber, and fluorescently labeled antibody is introduced and allowed to bind to equilibrium. The antibody is then washed out, and dissociation is measured by loss of fluorescence signal over time (Fig 2B). To confirm that photobleaching does not contribute to loss of signal during antibody dissociation, we compared virus intensities following 60 seconds of continuous versus endpoint-only image collection. We do not observe a difference in fluorescence between these samples, indicating that loss of fluorescence is due to antibody dissociation (S1 Fig). The initial dissociation rate of the antibody provides a value for $k_{off}$ which we can

compare between different antibodies in both IgG1 and Fab formats. By using a continuous-time model in which each arm of an IgG is assumed to dissociate at the rate of the corresponding Fab, we fit the experimentally-determined $k_{off}$ values of the Fab and IgG pairs to calculate $k_x$ (S1 Fig, details provided in Methods).

## Antigen crosslinking can slow antibody dissociation by as much as two orders of magnitude

We measured IgG and Fab dissociation kinetics for S139/1 and C05 against both WSN33 and HK68 (Fig 2C). Because the rapid dissociation of the S139/1 Fab against WSN33 HA approaches the sampling frequency of our assay, our measured $k_{off}$ in this case likely underestimates the true dissociation rate; we therefore also compared an off-rate estimated from reported BLI data [11]. We find that for both antibodies, the crosslinking rate is faster than the monovalent off-rate by at least an order of magnitude, and thus significantly boosts the ability of the IgG form of the antibody to remain bound (Fig 2D). In general, avidity will enhance affinity when new crosslinks form at a timescale similar to or faster than the monovalent dissociation rate (S1 Fig). Although antibody loading at very high concentrations could favor monovalent over bivalent attachment and lead to initial rapid dissociation kinetics, our measured dissociation rates are insensitive to the loading concentration of the antibody within the range that we test (S1 Fig). Avidity of these antibodies results in a decrease in dissociation rate of up to 150-fold, for S139/1 against HK68 HA. Although the S139/1 Fab dissociates twice as fast from HK68 HA as the C05 Fab, it's higher rate of crosslinking leads to slower dissociation when the two antibodies are compared as IgGs. This enhancement in effective affinity by multiple orders of magnitude parallels previous measurements for these and other antibodies [13,23].

## Decreasing HA surface density 25% through introduction of an HA decoy does not affect antibody binding avidity

The ability of an antibody to bind bivalently to an antigenic surface will depend on the availability of antigens that are suitably positioned, and should be reduced if HA density is sufficiently low. To experimentally reduce the HA surface density, we expressed a fluorescent HA decoy that competes with wildtype HA for packaging into virions in infected cells (Fig 3A). The decoy is comprised of the cytoplasmic tail and transmembrane domain of H1 HA, while the HA ectodomain is replaced by the GCN4-pII trimerization domain [30] fused to the fluorescent protein super-ecliptic pHluorin (SEP) [31]. To prevent co-trimerization of the wildtype HA and the decoy ('SEP-HA') through interactions between their transmembrane domains, we found that it is important that the SEP-HA transmembrane domain be derived from a different subtype than the virus used for infection; thus, for these experiments, we created an MDCK stable cell line expressing SEP-HA derived from WSN33 and infected with HK68 virus. This approach results in viruses that are morphologically similar to wildtype, albeit with a higher number of total particles released over the course of infection (S2 Fig). Virus collected from these infections is then used to determine the reduction of HA density that occurs due to competition between wildtype and SEP-HA, and the corresponding effect on antibody binding. Fluorescently labeled antibody is added to immobilized virus and allowed to bind to equilibrium, and the Fab and IgG steady-state binding is compared for the wild-type and mutant virions.

We reasoned that the monovalent binding of a Fab to an epitope at the apex of HA should be unaffected by changes in HA density. Accordingly, we used a C05 Fab fragment as a proxy for HA abundance to determine the average extent to which wildtype HA is diluted by the SEP-HA decoy on the surface of filamentous particles. We find that the SEP-HA virion population contains ~75% the amount of wildtype HA found in normal virions. This is somewhat higher than a separate estimate of HA density from Western blotting, where we find that the HA:M1 ratio in SEP-HA virus populations is 31% the value in wildtype populations (S2 Fig). The differences in these estimates may result from differences in the viral populations analyzed; in particular, a subset of particles in the SEP-HA samples lack wildtype HA, which are ignored in our filament analysis and binding assays and which would decrease the overall HA:M1 ratio as determined by western blot. At steady-state, both S139/1 and C05 IgG binding remains proportional to the HA content, at ~75% for SEP-HA virions relative to WT (Fig 3B). This indicates that the decrease in antibody binding is likely the direct result of the decrease in

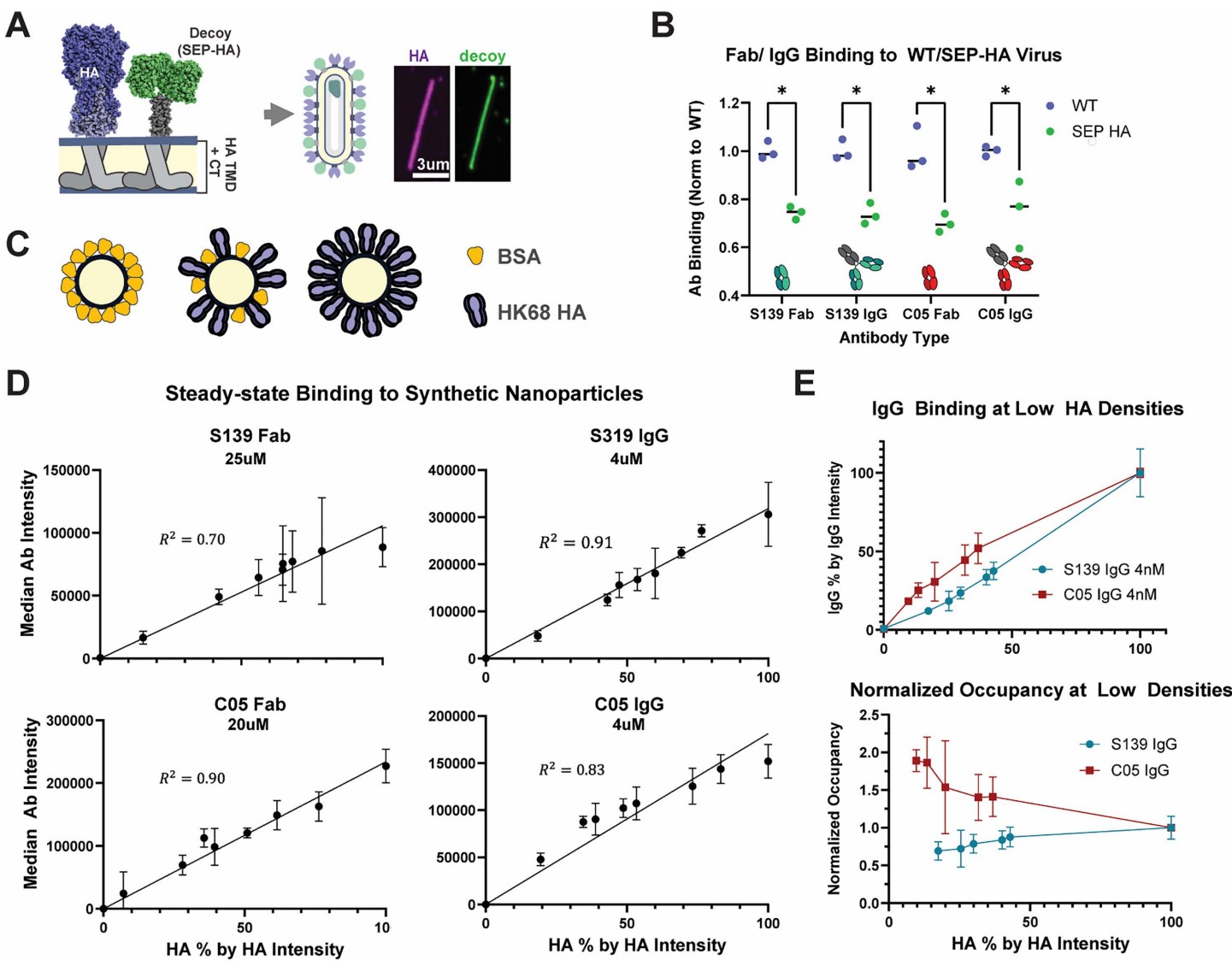

**Fig 3. S139/1 and C05 avidity persists at ten-fold reductions in HA density. (A)** Schematic and representative image of the fluorescent HA decoy used to reduce the HA density on the viral surface. In the panel to the left, HA is visualized using an scFv derived from FI6v3. The decoy (SEP-HA; shown in green in the panel to the right) contains the cytoplasmic tail and transmembrane domain of WSN33 HA, while the head domain is replaced by super-ecliptic pHluorin. **(B)** Steady-state binding of the antibodies to the WT and SEP-HA versions of the virus measured by fluorescence intensity. Quantification is performed on large filamentous virions to ensure high signal-to-noise. Measured intensities are normalized to the mean of the WT condition for each pair. P-values are determined by multiple unpaired t-tests for three replicates; significance is indicated by a p-value below 0.05. **(C)** Schematic of the experimental approach to decrease HA density using synthetic nanoparticles. C-terminally biotinylated HK68 HA ectodomain is bound to streptavidin-coated spheres and diluted with varying concentrations of biotinylated BSA to titrate the relative HA density. **(D)** Steady-state binding of Fab and IgG antibodies as a function of relative HA density. Relative HA density is determined by normalizing mean HA signal intensity to the 100% HA condition in each replicate. Three separate preparations of beads are measured for each condition. The R2 value of the linear regression for each antibody is shown correspondingly. **(E)** Comparison of S139/1 and C05 IgG binding at lower HA densities. Top: both HA and the antibody fluorescence intensities are also normalized to the mean 100% HA condition for each antibody for all three replicates. Bottom: antibody occupancy as a function of HA density. For each point, %IgG from the top plot is divided by the corresponding %HA.

available HA, rather than a change in bivalent binding; thus, a 25% decrease in surface density is not sufficient to diminish the avidity of either S139/1 or C05 IgG.

## Bivalent binding of S139/1 and C05 persists after ~10-fold reductions in HA surface densities

To probe the relationship between HA surface density and antibody binding more systematically, we used 200 nm streptavidin beads, which we can decorate with C-terminally biotinylated HA at different surface densities. Briefly, the beads are immobilized onto a coverslip, and purified HA from the HK68 strain that is fluorescently-labeled and biotinylated via a C-terminal ybbR tag [32] is introduced and allowed to bind to saturation. After washing to remove unbound HA, fluorescently-labeled antibody is added and allowed to reach equilibrium. To vary the HA surface density, biotinylated HA is mixed with varying concentrations of biotinylated BSA as a competitor (Fig 3C). In this way, we can measure steady-state antibody binding over a wide range of HA surface densities, with the highest density (100%) most closely mimicking a wild-type viral surface. We confirmed that surface densities of HAs on the beads are similar to those found on viruses by comparing binding of fluorescent C05 Fab to beads at 100% HA density relative to strain-matched virions. We note that the linker through which HA is anchored to the nanoparticle surface likely confers greater flexibility than the native HA membrane anchor (S2 Fig), although crowding by neighboring biotinylated proteins (BSA or HA) would be expected to suppress this effect. Accounting for differences in size [33], we estimate that HA density on the nanoparticle surface is ~85% that of the average HK68 virion (S2 Fig).

For both S139/1 and C05 Fab, binding increases linearly with HA density, as expected for a monovalent interaction dictated by absolute HA availability rather than density (Fig 3D). Interestingly, S139/1and C05 IgG also show similar trends. The amount of IgG bound appears directly proportional to the amount of HA available across a wide range of HA densities. At the lowest HA densities tested (~10%), we find that C05 IgG binds with highest efficiency; the ratio of bound IgG to HA ('normalized occupancy') is nearly double its value at the maximum HA density (Fig 3E). In contrast, for S139/1 the ratio of bound IgG to HA decreases modestly to ~75% when the HA density is reduced to ~15%. To confirm that these antibodies are binding bivalently at HA densities of 10%, we compared these results to results using Fabs. In contrast to the IgG, Fab binding measured at twice the molar concentration of the IgG (an equivalent concentration of binding sites) is nearly undetectable, confirming the IgG binding is not occurring through monovalent interactions (S2 Fig). The different binding behavior of C05 as compared to S139/1 IgG antibodies is particularly surprising, as it is opposite the trend we would expect if antigen density were the primary determinant of avidity.

## Tilting of HA about its membrane anchor contributes to C05 and S139/1 avidity

These findings suggest that antigen characteristics beyond surface density contribute to the efficiency of bivalent antibody binding, and that sensitivity to these characteristics may differ between antibodies. To better understand how structural features of antibody-HA complexes may contribute to avidity and dependence on antigen density, we used a structure-based model [15] to predict the orientations of adjacent HA trimers that could support bivalent binding by S139/1 and C05. By sampling different conformations of the antibody Fab domains and determining the resulting HA orientations from known structures, we obtain distributions for the linear distance between the base of the two HAs, as well as the angle between them (Fig 4A). This analysis suggests that while C05 binding prefers HA trimers that are further apart and oriented towards each other, S139/1 binding prefers trimers that are closer together and with an orientation that is closer to parallel. Of note, the preferred spacing for both antibodies are greater than the estimated distance (~10 nm) between proximal HAs on the native viral surface [9]. This suggests that the IgGs may bypass adjacent HAs to bind to secondary or even tertiary HA neighbors. Expanding this analysis to other known antibody-HA structures suggests that C05 is not unique (S3 Fig); for example, F045-092 also shows preference for a large inter-HA distance, and an even wider angle than C05, but has been shown to bind several orders of magnitude more strongly as IgG than as a Fab [34].

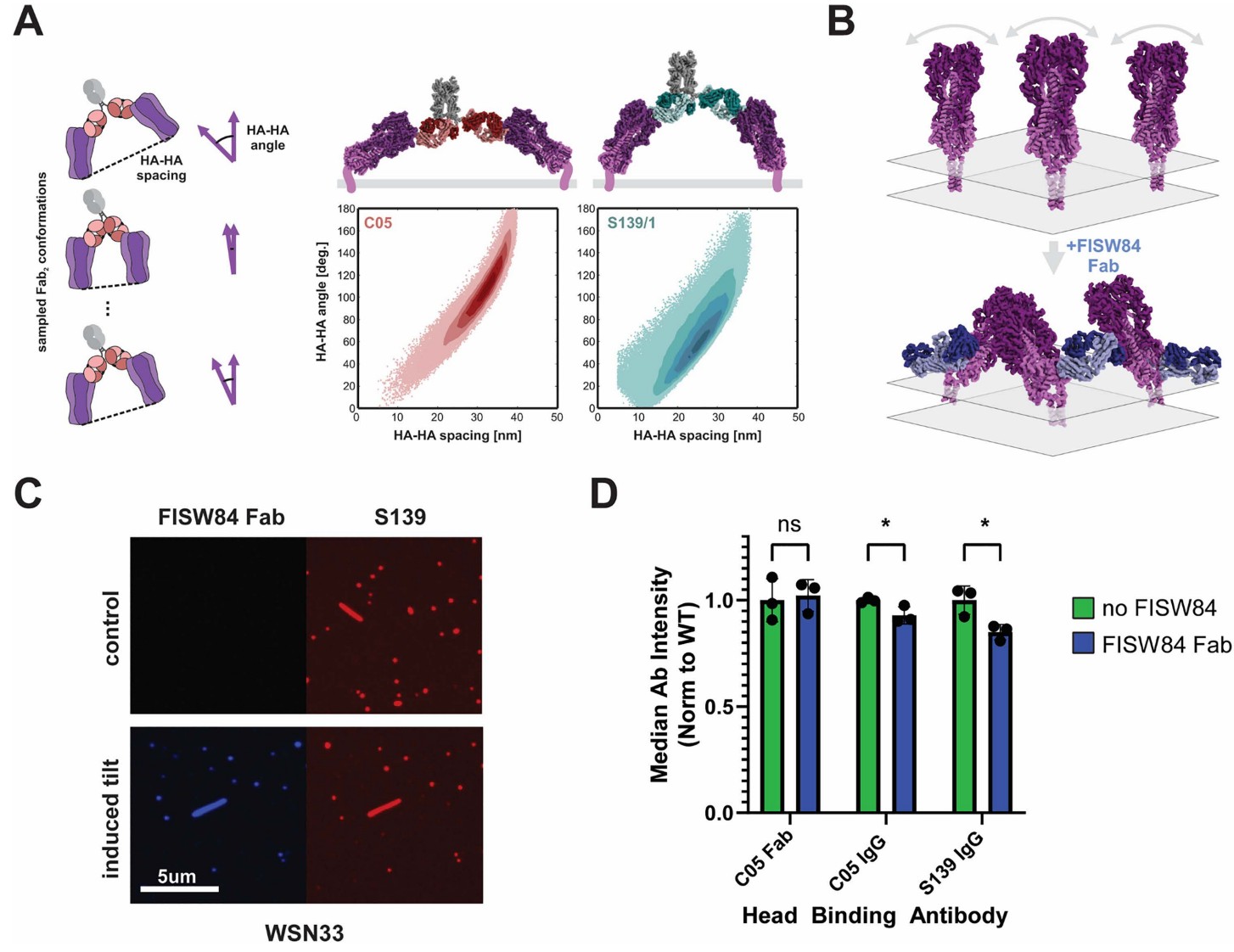

**Fig 4. Antibodies prefer different HA orientations for efficient crosslinking. (A)** Model to predict preferred antibody binding geometry. Structures of HA-Fab complexes are subjected to transformations reflecting the degrees of freedom of one Fab relative to the other. The distance between the base of the two bound HAs and the angle between them is determined for each sampled conformation. Plots in the lower right show angle and spacing distributions predicted for S139/1 and C05. Structural models above the plots are built by aligning HA-Fab structures (PDB IDs 4GMS and 4FQR) with human IgG1 (PDB ID 1HZH). **(B)** Schematic illustrating the proposed effect upon binding of an FISW84 Fab to HA (PDB IDs 6HJP, 6HJR). **(C)** Images of WSN33 bound by S139/1, +/- FISW84 Fab at 30nM. **(D)** Steady-state binding for the indicated antibodies. Analysis is performed on filamentous virions to ensure high signal-to-noise, and the median for each population is normalized to that of the corresponding wild-type condition. Three biological replicates are performed for each condition and compared by multiple unpaired t-tests; significance is indicated by a p-value below 0.05.

The HA ectodomain is connected to the transmembrane domain by a linker that permits extensive ectodomain tilting [35]. We hypothesized that this conformational flexibility may contribute to enhanced avidity by accommodating the structural preferences of different antibodies. If this is correct, we reasoned that avidity could be reduced by constraining tilting of the HA ectodomain. To test this hypothesis, we used FISW84, an antibody that binds to the HA anchor epitope [35] (Fig 4B). When one or two FISW84 Fabs binds to an HA trimer, tilting of the ectodomain will be constrained to the direction

opposite the Fab(s), reducing the likelihood that adjacent ectodomains will tilt towards each other at angles favorable for bivalent binding. Consistent with this hypothesis, viruses that are pre-bound by FISW84 Fab show reduced binding by S139/1 and C05 IgG, but not the C05 Fab (the affinity of the S139/1 Fab for A/WSN/1933 HA is not sufficiently strong for us to compare binding +/-FISW84) (Fig 4C & 4D). Thus, although the FISW84 Fab recognizes an epitope >10nm away from that recognized by either C05 or S139/1, it is able to modestly inhibit the binding of these antibodies, perhaps by perturbing the orientational dynamics of HA on the viral surface.

## Discussion

While the importance of antibody avidity in virus binding and neutralization is well established [15,23,36–38], the features of an antigenic surface that contribute to increased or decreased antibody avidity are not well understood. Using experimental approaches which directly perturb HA surface density and orientation, we are able to gain some insight into how these variables affect the avidity of two antibodies that bind to the HA head at different angles of approach – S139/1 and C05. We find that both antibodies are insensitive to at least a 10-fold decrease in HA surface density. This result is supported by structural predictions that suggest that these and other antibodies which bind to the HA head may rarely crosslink nearest neighbors within the viral membrane, binding instead to HAs whose anchors are separated by 20–30nm. Determining how these findings generalize to antibodies beyond the two tested here will require additional work. It is also important to note that we have focused here on IgG1 antibodies; differences in the hinge and valency of other antibody isotypes or subclasses are likely to be important [39,40].

In addition to HA density, our data and analysis suggest that the ability of HA to tilt about its membrane anchor may play an important role in supporting bivalent attachment by some antibodies. While we predict that S139/1 can bridge two HAs tilted ~30° towards each other with respect to the axis normal to the plane of the membrane, C05 favors an angle nearly twice as high (~60°). In comparison, neighboring HA trimers on the viral surface would be expected to splay away from each other at an angle of ~15° if they are oriented normal to the membrane [9]. This difference in preferred HA orientation may explain why S139/1 is more strongly affected by the presence of FISW84 (Fig 4), which we expect will bias HAs into a more tilted conformation. It may also explain why C05 shows higher binding occupancy at low HA densities in experiments with synthetic nanoparticles (Fig 3); lower HA densities may facilitate sampling of extreme angles that are obstructed when HA densities are high. Recent data has shown that antibodies bound to HA on the infected cell surface can inhibit viral assembly [15] and alter viral morphology [41], at presumably lower HA densities than those found on the viral surface. Direct measurements of HA conformational dynamics in the presence or absence of antibodies could provide more insight into the relationship between density and freedom of motion. In our experiments using streptavidin-coated nanoparticles, biotinylated HA would not be able to diffuse laterally, reflecting previous photobleaching data that indicates that long-range diffusion of HA and NA does not occur on the viral surface [42]. Short-range diffusion, while not observed, remains a possibility; if present, we expect that this would further facilitate bivalent binding at low HA densities by allowing antibodies to capture HAs that transiently come in close proximity.

The density of glycoproteins on the viral surface poses a potential evolutionary tradeoff; while high densities facilitate viral attachment to the cell surface, they may also make the virus more susceptible to neutralization by high-avidity antibodies. There is some evidence that different viruses have arrived at different evolutionary solutions to this problem. HIV-1 virions have only a few copies of the Env protein per virus – sufficient for binding the viral receptor through high affinity, but at a distance that is too far apart for an IgG to bind bivalently [43,44]. Without enhanced binding through avidity, neutralization must be achieved primarily through antibodies with a high monovalent affinity [36,45]. Influenza is at the opposite extreme, with neighboring glycoproteins separated by ~10nm from each other. Our results suggest that reducing HA surface densities by up to 10-fold would have minimal effect on the avidity of the antibodies tested here. If this is the case, there would be little evolutionary pressure towards lower HA surface densities, which could reduce viral attachment in cases where high-affinity sialic acid receptors are limited without reducing the occupancy of some avidity-dependent

antibodies. Besides HA density, our results and analysis suggest an important role for HA flexibility in increasing antibody avidity. This, too, could be important for viral fitness, for example, by allowing the HA ectodomain to pivot about its membrane anchor during the foldback step of membrane fusion, or to better engage cellular receptors. Genetic approaches to tune HA flexibility and incorporation into budding virions beyond what we demonstrate here will be critical in testing these predictions.

## Materials and methods

### Cell and virus cultivation

MDCK-II and HEK-293T cell lines used in the study were purchased from ATCC as authenticated cell lines (STR profiling). They were grown under standard conditions (37°C and 5% $CO_2$) and cultured with cell growth medium containing Dulbecco's modified Eagle's medium (DMEM; Gibco), 10% fetal bovine serum (FBS; Gibco) and 1 × antibiotic-antimycotic (Corning).

Standard reverse genetics techniques were used for virus rescue of the WSN33 and HK68 strains [46]. Briefly, co-cultures of HEK-293T and MDCK-II were transfected with plasmids containing each of the eight vRNA segments flanked by bidirectional promoters. The virus was harvested approximately 48 hours after transfection and plaque purified. For virus expansion, the plaques were passaged at a low MOI (~0.001) in MDCK-II cells, in a virus growth medium containing Opti-MEM (Gibco), 2.5 mg/mL bovine serum albumin (Sigma-Aldrich), 1 µg/mL L-(tosylamido-2-phenyl ethyl) chloromethyl ketone (TPCK)-treated trypsin (Thermo Scientific Pierce), and 1 × antibiotic-antimycotic (Corning). The viral stocks were further expanded for experiments by passaging at low MOI in MDCK-II cells. Here, a version of WSN33 with filamentous morphology was used to match the filamentous morphology observed in HK68 virions. To obtain this phenotype, the WSN M1 sequence was replaced by that of A/Udorn/1972 (genetically similar to HK68), as described previously [47].

### Protein purification and labeling

Sequences for the VH and VL regions of the antibodies of interest were obtained from deposited sequences and cloned into expression vectors to make either full-length human IgG1 antibodies or Fab fragments. The heavy chain sequences were modified with a C-terminal ybbR tag for enzymatic labeling and purified as described [48]. Briefly, full-length antibodies were purified using protein A agarose beads (Thermo Scientific Pierce), while $His_6$-tagged Fab fragments were affinity purified using Ni-NTA Agarose Beads (Thermo Scientific Pierce). Adherent HEK293T cells were allowed to grow to ~90% confluency, transfected with the heavy and light chain plasmids, and allowed to express for 7 days. During this time, the cells were cultured in Opti-MEM with 1 × antibiotic-antimycotic and 2% FBS for Fab expression, and without FBS for IgG expression.

Recombinant HA for experiments with beads was produced by transfecting HEK293T cells with a pCAGGS expression vector containing the sequence for the HK68 HA ectodomain, followed by a foldon, $His_6$-tag for affinity purification, and ybbR tag for enzymatic biotinylation using Sfp synthase [32]. After expressing for five days, HA was purified from the supernatant with Ni-NTA Agarose Beads (Thermo Scientific Pierce), and enzymatically biotinylated overnight at 4°C with Sfp. The HA was then labeled using Sulfo-Cy5 NHS dye (Lumiprobe) to achieve a ratio of approximately one dye molecule per HA trimer. Sequences of constructs used in this work are given in S1 Text.

### Imaging flow chamber construction and functionalization

Imaging flow chambers were built with channels of 1mm width and 0.17mm height. The chambers were constructed by securing no. 1.5 thickness coverslips to acrylic covers with a double-sided adhesive (3M). The 1/16" acrylic backing was laser cut to form wells, and the adhesive was vinyl cut to shape the channels.

The coverslips were functionalized via a series of incubation steps carried out at room temperature. First, 90 µg/mL of BSA-biotin was flowed into the chambers and allowed to adsorb to the coverslip for two hours. Remaining BSA-biotin was

washed with an excess of PBS, then incubated with 25 µg/mL streptavidin (Invitrogen) in PBS for 1 hour. Flow chambers were washed again with excess PBS and subsequently incubated with 25 µg/mL biotinylated *Erythrina cristagalli* lectin (ECL; Vector Laboratories) for 2 hours. ECL is used to capture viruses on the coverslip. The channels were washed a final time with PBS before introducing virus for imaging.

## Antibody dissociation assay

Virions were immobilized in the functionalized flow chambers. The fluorescently labeled antibody of interest was added, and binding was allowed to reach equilibrium. The imaging chamber was then set on a Nikon TI2 confocal microscopy system, using a 60×, 1.40-NA objective. The antibody was washed out with PBS, and dissociation was observed through loss of fluorescence signal. Images were acquired in a timelapse, with frame rate set relative to the expected dissociation rate to minimize photobleaching.

Antibodies used for dissociation measurements were labeled with Alexa Fluor 555, selected for its brightness and excellent photostability. To confirm that the loss of signal in dissociation measurements was not due to photobleaching, we compared loss of antibody signal under two conditions: (1) Dissociation only: images were collected at 0 and 60s after removal of antibody; and (2) Dissociation and photobleaching: images were collected at 1s intervals for 60 seconds. The number of acquisitions did not affect the endpoint intensities (S1 Fig).

## SEP-HA cells lines and virus

To create the SEP-HA expressing cell line, lentivirus was produced by transfecting the packaging and envelope vectors pCMV8.91 and pMD2.G, along with transfer vector pHR-SIN into HEK29T cells. The SEP-HA construct consisted of SEP, a GCN4-pII trimerization domain, and the WSN33 HA transmembrane domain/cytoplasmic tail, followed by an internal ribosome entry site (IRES) for expression of puroR. MDCK-II cells were infected with this lentivirus and selected with puromycin to create a cell line that constitutively expresses SEP-HA. To achieve uniform expression of SEP-HA, we performed a clonal expansion from single cells in 96-well plates and selected the highest expressing clone (as determined by cell surface SEP signal) for subsequent experiments.

SEP-HA-containing influenza virus was obtained by infecting this cell line with HK68 at an MOI of ~1, while wild type virus was collected from the parental MDCK cell line for comparison. Virus was collected 18 hours post infection and immobilized in flow chambers. Fluorescent antibody was washed into the channels and allowed to bind to equilibrium. Signal of both the SEP and antibody fluorophore were collected at steady-state following incubation for 3–4 hours. Filamentous particles were segmented during image processing to ensure high signal-to-noise in data analysis. The mean intensity per pixel was measured for each filamentous virion (defined as a particle with a major axis at least four times as long as the minor axis).

To quantify the relative amounts of authentic HA present in each sample, western blotting was performed using virus-containing supernatant collected from wild-type and SEP-HA-expressing cells 18 hours post-infection. Supernatants were clarified and then concentrated by centrifugation at 22,000×g. Three technical replicates were run simultaneously on the same gel. Samples were probed with an HA-tag antibody (Thermo Fisher Scientific 26183) to detect HK68 HA and an anti-M1 antibody (GA2B, Santa Cruz Biotechnology). HA and M1 primary antibodies were detected by orthogonally labeled fluorescent secondary antibodies using the LI-COR Odyssey CLx.

## Synthetic nanoparticles

C-terminally biotinylated HA was bound to streptavidin-coated beads of 200nm diameter (Bangs Laboratories) to mimic the viral surface. A glass-bottom 96 well plate was cleaned by incubating wells with 3M NaOH for 30 minutes and washing thoroughly with PBS. The plate was then incubated with BSA-biotin at room temperature for 1 hour to functionalize the surface. The streptavidin beads were diluted into PBS with 0.05% tween (PBST) and added to the wells for 15 minutes.

Relative HA density on the surface of beads was titrated by mixing biotinylated HA with varying concentrations of biotinylated BSA. The 100% HA solution contained 16nM of the purified HA, while the 0% HA solution contained 150nM BSA-biotin; mixtures of these solutions were prepared to achieve the intermediate concentrations shown in Fig 3. Each mixture was added to a well and allowed to bind to the beads at 4°C overnight. The excess protein was washed thoroughly with PBS, and fluorescently labeled antibodies were added to the wells and allowed to bind for 4 hours. Images were collected using a Nikon TI2 confocal microscope with a 60×, 1.40-NA objective. Relative HA density and antibody binding were determined by normalizing signals to HA and antibody fluorescence intensity respectively in the 100% HA condition.

## Competition assay with FISW84

WSN33 virus was produced by infecting MDCK cells at an MOI of ~1 and collecting the virus-containing supernatant at 18h post infection. Virus was immobilized into functionalized flow chambers. To induce HA tilt, a fluorescent Fab derived from the antibody FISW84 was added at 30nM and allowed to bind for 1 hour; the Fab form was used so as not to confound results with any potential steric hindrance from the Fc. The fluorescent antibody of interest was then added (along with 30nM of FISW84 for the tilt-induced condition). As before, filamentous particles were segmented during image processing to ensure high signal-to-noise in data analysis.

## Determining antibody crosslinking rates from experimental data

For the crosslinking model shown schematically in Fig 2A, the evolution of singly-bound ($A_1$) and doubly-bound antibodies ($A_2$) over time is given by the equations:

$$\frac{dA_1}{dt} = k_{on}\left[A_0\right]\left(1 - A_1 - 2A_2\right) - k_{off}A_1 - k_x\left(1 - A_1 - 2A_2\right)A_1 + k_{off}A_2 \tag{1}$$

$$\frac{dA_2}{dt} = k_x\left(1 - A_1 - 2A_2\right)A_1 - k_{off}A_2 \tag{2}$$

Here, $A_1$ and $A_2$ are normalized by the total number of HAs in the system, $[A_0]$ denotes the molar concentration of IgG in solution (assumed constant), $k_{on}$ denotes the association rate for the IgG antibody (taken as twice the association rate of the Fab), $k_{off}$ denotes the monovalent (Fab) dissociation rate, and $k_x$ denotes the crosslinking rate. To determine values for $k_x$ from measured IgG and Fab dissociation rates, we numerically integrate $A_1$ and $A_2$ and fit the sum (corresponding to the total bound antibody, the quantity that is captured by our measurements) to a kinetic model without crosslinking:

$$\frac{dA_{IgG}}{dt} = k_{on}\left[A_0\right]\left(N - 2A_{IgG}\right) - k_{off}^{IgG}A_{IgG} \tag{3}$$

Here, $k_{off}^{IgG}$ corresponds to the apparent dissociation rate measured for IgG antibodies (Fig 2D), and we assume that all antibodies attach bivalently. For the high-avidity antibodies S139/1 and C05, this assumption is reasonable; for the concentrations of antibodies we use, our kinetic parameters suggest that $A_2$ is typically ~10–100-fold greater than $A_1$. For fitting the crosslinking rate $k_x$, we focus only on the dissociation phase, initializing the system using $A_2(t=0) = A_{IgG}(t=0)$ = 0.2 and $A_1(t=0)$ = 0. This fitting procedure is used to determine the values for $k_x$ reported in Fig 2D. It is important to note that as the initial value of $A_2$ approaches its maximum value of 0.5, fitting becomes poor. Under these conditions, as the first doubly-bound antibodies dissociate into singly-bound antibodies, they are unable to efficiently re-form crosslinks because the majority of HAs remain occupied. This leads to an initial rapid dissociation phase which we do not observe in our experiments. This is a limitation of the continuum model, which does not preserve spatial information regarding the location of free HAs relative to antibodies that had previously been bound to them.

Solving (1) and (2) for the total fraction of bound HAs gives:

$$A_1 + 2A_2 = 1 - \frac{k_{on}[A_0]}{4k_x\varphi}\left(1 - \frac{1}{\varphi}\right)\left[\left(1 + \frac{8k_x\varphi^2}{k_{on}[A_0]}\right)^{\frac{1}{2}} - 1\right]$$

(4)

where

$$A_1 = \frac{k_{on}[A_0]}{4k_x\varphi}\left[\left(1 + \frac{8k_x\varphi^2}{k_{on}[A_0]}\right)^{\frac{1}{2}} - 1\right]$$

$$A_2 = \frac{1}{2} - \frac{k_{on}[A_0]}{8k_x\varphi^2}\left[\left(1 + \frac{8k_x\varphi^2}{k_{on}[A_0]}\right)^{1/2} - 1\right]$$

$$\varphi \equiv \frac{k_{on}[A_0]}{k_{on}[A_0] + k_{off}}$$

Setting (4) equal to ½ and solving for the antibody concentration $[A_0]$ allows us to determine the fold decrease in $K_D$ for an IgG antibody relative to the Fab as a function of the crosslinking rate $k_x$ and other kinetic parameters. These results are plotted in S1 Fig. Matlab code for fitting and analysis is available on Github (https://github.com/mvahey/benegal2024).

### Geometric Model for Preferred Crosslinking Geometry

To model the preferred crosslinking geometry for each antibody, atomic coordinates of Fab fragments bound to their HA epitope were acquired from the Protein Data Bank and aligned. Potential configurations for the second Fab arm were sampled by translation and rotation transformations, with the range of motion for each degree of freedom informed by previously recorded behavior of antibody flexibility [49,50]. Conformations were sampled for Fab arm twisting of $\psi \pm 60°$, rotation along the Fc axis of $\phi = 180° \pm 30°$, and the Fab domain angle at $\theta = 60° \pm 30°$, with "±" indicating the standard deviations of the distribution from which each angle was sampled. An extra "wobble" parameter was included in order to allow for the breaking of symmetry. A schematic is shown in S3 Fig.

For each sampled configuration, the scalar product of the axial vectors that extend from the bottom to top of the two HAs was measured. From this information, the angle between the two HAs, as well as the distance between their bases, were recorded. For S139/1 and C05, the distribution of each corresponding angle/distance pair is shown in Fig 4A. The analysis was extended to a broader range of structures of HA-antibody complexes from the PDB. Panel B in S3 Fig shows the most frequent inter-HA spacing and angle sampled by each particular antibody, while panel C maps the structure of an HA monomer with each residue colored according to the frequency of contacts by antibodies in the corresponding groups from the plot in panel B.

### Statistics, replicates, and software

Statistical analysis was performed using GraphPad Prism 10 and MATLAB. The statistical tests and the number of replicates used in specific cases are described in the figure legends. No statistical methods were used to predetermine sample size.

### Supporting information

**S1 Fig. Characterization of antibody binding and crosslinking kinetics.** (A) To control for photobleaching, the loss of fluorescence signal was imaged under two conditions: 1) dissociation only; an image was collected at 0s and one at 60s and 2) dissociation and photobleaching; an image was collected at a rate of 1 frame per second for 60 seconds. The

difference between the endpoint intensities from both conditions is not statistically significant. (B) To account for any differences in weighting the average, the normalized dissociation curves were averaged in two ways: 1) by taking the average with each virion equally weighted, and 2) with the entire field of view masked and normalized together. (C)To fit the cross-linking rate ($k_x$), the experimentally obtained $k_{off}$ values from each Fab and IgG pair were mapped to the simulation results. For a given Fab $k_{off}$ value, we iterate through a series of $k_x$ values and determine the best fit to the dissociation curve for the corresponding IgG. (D) The fold difference in effective $k_{off}$ is shown as a function of $k_x$ for a few given Fab $k_{off}$ values. (E) Intensities of bound antibodies prior to measuring dissociation. Data is from the experiments shown normalized in Fig 1C. The concentration for each antibody is shown in blue or green for the higher and lower concentrations, respectively (F) Normalized dissociation curves for the conditions plotted in D.
(TIF)

**S2 Fig. Determining the effects of reduced antigen density on antibody avidity.** Comparisons of wildtype and SEP-HA virus populations based upon (A) confocal microscopy analysis of the total particle count and percentage of the virion population comprised of filamentous particles; and (B) filament length distributions, and (C) a quantification of Western blot intensity values of relative HA signal normalized by M1 in each supernatant population; HA and M1 are quantified on the same gel by different antibodies. (D) Comparison of the structures of native HA (PDB ID 6HJQ) and the recombinant HA with trimerization domain and linker used in synthetic nanoparticle experiments (AlphaFold2). (E) Calibration of HA density on streptavidin nanoparticles relative to native virions. Plot to the left quantifies the fluorescent intensity of labeled S139/1 IgG bound to nanoparticles (incubated with biotinylated HA) and viruses. Schematic to the right shows relative sizes of nanoparticles and virions used to calculate relative HA densities. The corresponding scaling between the size of an average spherical virion to a nanoparticle is shown. (F) Quantification of HA intensities (left plot) and antibody intensities (right plot) for beads with saturating densities ('100%') of HA, incubated with S139/1 IgG or S139/1 Fab.
(TIF)

**S3 Fig. Modeling the geometric preferences of HA-specific monoclonal antibodies.** (A) Schematic representing the geometric binding model. For a given HA-Fab interaction, the first Fab arm (Fab1) is aligned to its binding site on HA. The position of the second arm is simulated by sampling conformations about the degrees of freedom indicated. (B) Predictions from the structure-based model from Fig 4A, extended to structures of HA-antibody complexes from the PDB. Each point represents the most frequent inter-HA spacing and angle samples by a particular antibody. S139/1, C05, and F045-092, another high-avidity antibody, are highlighted. Stem-binding antibodies are indicated by a black outline. (C) Structure of an HA monomer with each residue colored according to the frequency of contacts by antibodies in the corresponding groups from the plot in *A*. Contacts are defined as HA residues within 0.8 nm of any residue from the antibody heavy or light chain.
(TIF)

**S1 Data. Raw data used to generate article figures.**
(XLSX)

**S2 Data. Raw data used to generate supporting information figures.**
(XLSX)

**S1 Text. Sequences of the antibodies and antibody fragments used in binding experiments.**
(DOCX)

## Author contributions

**Conceptualization:** Ananya N. Benegal, Michael D Vahey.

**Data curation:** Ananya N. Benegal.

**Formal analysis:** Ananya N. Benegal, Michael D Vahey.

**Funding acquisition:** Michael D Vahey.

**Investigation:** Ananya N. Benegal, Kaitlyn Ho, Giselle Groff.

**Methodology:** Ananya N. Benegal.

**Project administration:** Michael D Vahey.

**Resources:** Yuanyuan He, Zijian Guo, Michael D Vahey.

**Writing – original draft:** Ananya N. Benegal, Michael D Vahey.

**Writing – review & editing:** Ananya N. Benegal, Michael D Vahey.

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
