## [Decision Letter · Decision Letter 0]

30 Sep 2025

PPATHOGENS-D-25-01596

Antigen flexibility supports the avidity of hemagglutinin-specific antibodies at low antigen densities

PLOS Pathogens

Dear Dr. Vahey,

Thank you for submitting your manuscript to PLOS Pathogens. After careful consideration, we feel that it has merit but does not fully meet PLOS Pathogens's publication criteria as it currently stands. Therefore, we invite you to submit a revised version of the manuscript that addresses the points raised during the review process.

Please submit your revised manuscript within 30 days Nov 29 2025 11:59PM. If you will need more time than this to complete your revisions, please reply to this message or contact the journal office at plospathogens@plos.org. Please include the following items when submitting your revised manuscript:

We look forward to receiving your revised manuscript.

Kind regards,

Alan G. Goodman

Academic Editor

PLOS Pathogens

Thomas Hoenen

Section Editor

PLOS Pathogens

Sumita Bhaduri-McIntosh

Editor-in-Chief

PLOS Pathogens

orcid.org/0000-0003-2946-9497

Michael Malim

Editor-in-Chief

PLOS Pathogens

orcid.org/0000-0002-7699-2064

**Additional Editor Comments:**

Your revised manuscript has been positively reviewed by the three reviewers. Reviewer #1 has remaining minor concerns that should be addressed before moving forward with your manuscript. In Figure S3, please highlight the data points in cluster "i" that correspond to the stem antibodies. Additionally, the dataset should be included as a supplementary table listing each antibody with its most frequent inter-HA spacing and angle. The Github link needs correcting too.

**Journal Requirements:**

https://journals.plos.org/plospathogens/s/submission-guidelines#loc-parts-of-a-submission

5) Please note that your Data Availability Statement is currently missing the DOI/accession number of each dataset OR a direct link to access each dataset. If your manuscript is accepted for publication, you will be asked to provide these details on a very short timeline. We therefore suggest that you provide this information now, though we will not hold up the peer review process if you are unable.

2) If any authors received a salary from any of your funders, please state which authors and which funders..

7) Your current Financial Disclosure states, "Yes ↳ Please add funding details. National Science Foundation (NSF):Ananya Benegal,Michael Vahey 2238165; HHS | National Institutes of Health (NIH):Yuanyuan D He,Zijian Guo,Michael Vahey AI171445 ↳ Please select the country of your main research funder (please select carefully as in some cases this is used in fee calculation). UNITED STATES - US".

However, your funding information on the submission form indicates different funders.

Please indicate by return email the full and correct funding information for your study and confirm the order in which funding contributions should appear. Please be sure to indicate whether the funders played any role in the study design, data collection and analysis, decision to publish, or preparation of the manuscript.

8) Please send a completed 'Competing Interests' statement, including any COIs declared by your co-authors. If you have no competing interests to declare, please state "The authors have declared that no competing interests exist". Otherwise please declare all competing interests beginning with the statement "I have read the journal's policy and the authors of this manuscript have the following competing interests"

**Reviewers' Comments:**

Reviewer's Responses to Questions

**Part I - Summary**

Reviewer #1: I previously reviewed this manuscript on Review Commons. The authors have done an excellent job of addressing the concerns raised in the previous round of review. The manuscript is much improved, and I recommend it for publication. I have no further comments besides a very minor, optional point for the modeling data in Fig. S3 (please see below).

Reviewer #2: The authors of the PLoS Pathogens manuscript "Antigen flexibility supports the avidity of hemagglutinin-specific antibodies at low antigen densities" present their recent work evaluating antibody interactions with influenza virus proteins. Specifically, they show that the density of the hemagglutinin protein on the surface of the virus impacts antibody binding. Their results show that bivalent antibody binding persists down to one-tenth the density seen on virus particles and that reduced hemagglutinin expression can actually increase antibody occupancy. When hemagglutinin was locked at a fixed angle, the authors were able to show that hemagglutinin flexibility impacts antibody avidity, thereby concluding that the binding of neutralizing antibodies is impacted by surface antigen expression. The authors have responded to comments provided through Review Commons, and their edits and added controls have been incorporated into the manuscript that is under consideration. Below are my thoughts regarding the manuscript and the response to previous comment.

Reviewer #3: In “Antigen flexibility supports the avidity of hemagglutinin-specific antibodies at low antigen

densities”, Benegal et al. develop a microscopy-based assay to measure dissociation of HA head-

binding antibodies from intact virions. This assay allows the authors to explore the contribution of

IgG bivalent avidity to antibody interaction with native virions, which is not accessible using other

methods such as BLI. Using this assay, the authors further explore the effect of HA density on IgG

avidity with engineered low-HA virions and then with artificial HA-coated microspheres. In addition

to measuring antibody dissociation, the authors perform structural analyses to predict the

conformational preferences of many HA IgGs from published structures. The authors conclude that

low HA densities (down to ~10%) still support high avidity binding for the 2 IgGs tested, and thus

there would be little evolutionary pressure for IAV to reduce the HA density as a strategy to evade

immune recognition.

**Part II – Major Issues: Key Experiments Required for Acceptance**

Please use this section to detail the key new experiments or modifications of existing experiments that should be absolutely required to validate study conclusions.required to validate study conclusions.

Reviewer #1: (No Response)

Reviewer #2: Authors have clarified comments related to virus selection and morphology, presentation of findings, clarification of key points, and issues related to photobleaching.

Reviewer #3: All my previous comments and those of other reviewers were adequately addressed.

**Part III – Minor Issues: Editorial and Data Presentation Modifications**

Reviewer #1: Regarding Figure S3, I noticed that cluster 'i' appears to contain nearly all of the HA stem antibodies, plus a few from the head region. Would it be possibe to highlight which data points within this cluster correspond specifically to the stem antibodies?

Furthermore, while this modeling is in the supplemental material, the dataset itself is quite valuable. It would be a great benefit to the field if the authors could make this data available to readers, perhaps in a new supplementary table listing each antibody with its most frequent inter-HA spacing and angle and/or in the authors' GitHub link.

Of note, it also looks like the authors' Github link is still not working, which they may want to fix for the final version of the manuscript.

Reviewer #2: None noted, the data are clearly presented.

Reviewer #3: All my previous comments and those of other reviewers were adequately addressed.

PLOS authors have the option to publish the peer review history of their article (what does this mean? ). If published, this will include your full peer review and any attached files.). If published, this will include your full peer review and any attached files.

**Do you want your identity to be public for this peer review?** For information about this choice, including consent withdrawal, please see our For information about this choice, including consent withdrawal, please see our Privacy Policy ..

Reviewer #1: No

Reviewer #2: No

Reviewer #3: No

**Figure resubmission:**
---

## [Editor Report · Decision Letter 1]

1 Jan 2026

Dear Assistant Professor Vahey,

We are pleased to inform you that your manuscript 'Antigen flexibility supports the avidity of hemagglutinin-specific antibodies at low antigen densities' has been provisionally accepted for publication in PLOS Pathogens.

Best regards,

Alan G. Goodman

Academic Editor

PLOS Pathogens

Thomas Hoenen

Section Editor

PLOS Pathogens

Sumita Bhaduri-McIntosh

Editor-in-Chief

PLOS Pathogens

orcid.org/0000-0003-2946-9497

Michael Malim

Editor-in-Chief

PLOS Pathogens

orcid.org/0000-0002-7699-2064

Thank you for revising your manuscript based on the reviewers' remaining minor concerns. I am pleased to recommend acceptance of your manuscript and proceed with the publication process at Plos Pathogens.
---

## [Editor Report · Acceptance letter]

Dear Assistant Professor Vahey,

We are delighted to inform you that your manuscript, "Antigen flexibility supports the avidity of hemagglutinin-specific antibodies at low antigen densities," has been formally accepted for publication in PLOS Pathogens.

Best regards,

Sumita Bhaduri-McIntosh

Editor-in-Chief

PLOS Pathogens

orcid.org/0000-0003-2946-9497

Michael Malim

Editor-in-Chief

PLOS Pathogens

orcid.org/0000-0002-7699-2064